# Original Leaf Colonisers Shape Fungal Decomposer Communities of *Phragmites australis* in Intermittent Habitats

**DOI:** 10.3390/jof8030284

**Published:** 2022-03-10

**Authors:** Matevž Likar, Mateja Grašič, Blaž Stres, Marjana Regvar, Alenka Gaberščik

**Affiliations:** 1Department of Biology, Biotechnical Faculty, University of Ljubljana, 1000 Ljubljana, Slovenia; mateja.grasic@bf.uni-lj.si (M.G.); marjana.regvar@bf.uni-lj.si (M.R.); alenka.gaberscik@bf.uni-lj.si (A.G.); 2Department of Animal Science, Biotechnical Faculty, University of Ljubljana, 1000 Ljubljana, Slovenia; blaz.stres@bf.uni-lj.si

**Keywords:** community networks, fungal communities, keystone species, litter decomposition, *Phragmites australis*, water regime

## Abstract

Common reed (*Phragmites australis*) has high biomass production and is primarily subjected to decomposition processes affected by multiple factors. To predict litter decomposition dynamics in intermittent lakes, it is critical to understand how communities of fungi, as the primary decomposers, form under different habitat conditions. This study reports the shotgun metagenomic sequencing of the initial fungal communities on common reed leaves decomposing under different environmental conditions. We demonstrate that a complex network of fungi forms already on the plant persists into the decomposition phase. *Phragmites australis* leaves contained at least five fungal phyla, with abundant Ascomycota (95.7%) and Basidiomycota (4.1%), identified as saprotrophs (48.6%), pathotrophs (22.5%), and symbiotrophs (12.6%). Most of the correlations between fungi in fresh and decomposing leaves were identified as co-occurrences (positive correlations). The geographic source of litter and leaf age did not affect the structure and diversity of fungal communities. Keystone taxa were mostly moisture-sensitive. Our results suggest that habitat has a strong effect on the formation of the fungal communities through keystone taxa. Nevertheless, it can also alter the proportions of individual fungal groups in the community through indirect effects on competition between the fungal taxa and their exploitation of favourable conditions.

## 1. Introduction

Wetlands are highly productive ecosystems, especially those colonised by helophytes [1]. The level of herbivory in such wetlands is relatively low, thus the majority of produced biomass is subjected to the decomposition processes [2]. *Phragmites australis* is a highly productive cosmopolitan species that forms extensive monoculture stands in many wetlands [1] and contributes significantly to plant litter production in these ecosystems [3,4]. Leaf litter has multiple functions in the ecosystem. It presents a source of organic material and nutrients, prevents soil erosion and compaction, and creates favourable conditions for a variety of organisms [5].

In *P. australis*, microbial colonisation of plant material starts when plants are still upright [6,7]. Significant changes in microbial colonisation and their activity occur after the collapsing of plants [8], when plant material accumulates on the ecosystem floor. The effect of these standing reed shots falling into the litter layer on the microbial community depends on the duration of their standing decay and the time of the year when the process is happening [6]. In wetlands, this litter layer may be flooded or dry. Flooding has a strong effect on the formation of microbial communities in different ways [9]. The presence of water in the habitat indirectly affects the decomposition rate by changes in oxygen availability, moisture, and temperatures [10]. Furthermore, it also affects biotic interactions that are important determinants of microbial community development [11,12,13]. Along with the presence of water, the frequency and the duration of the submergence periods present further important variables shaping microbial communities [12,14]. Likar et al. [15] showed pronounced differences in fungal diversity on *P. australis* litter, with the lowest diversity developed on the litter decomposing at the location with a changing water level. Changes in fungal communities directly affect the decomposition process, as it depends on species richness and the development of the microbial community on the litter [16].

The relationship between communities of decomposers and their contribution to litter decay is still poorly understood. Established theories of decomposition characterise fungi as the main microbes in the primary decomposition of large pieces of organic matter [17] as they produce a wide range of extracellular enzymes involved in the breakdown of the lignocellulose matrix [5]. In contrast, bacteria degrade detrital particulate and dissolved organic matter [18]. On the contrary to this view, some recent observations revealed that both bacteria and fungi are important players in the decomposition process in all stages of decay [19]. The resource partitioning theory interprets the interspecies relationships of a microbial community during decomposition and the high efficiency of decay driven by complementarity in resource use among the dominant species of a community [20,21]. Although, as described above, much effort has been made in studying litter decomposition, the mechanisms behind shifts of a microbial community during decomposition and the contributions of different microbial decomposers involved in litter decay are still obscure.

To predict *P**. australis* litter decomposition dynamics in intermittent lakes, it is critical to understand how the fungal communities present in fresh leaves change in terms of their richness and diversity under different habitat conditions. Furthermore, it is of great interest to evaluate the importance of the plant phytobiome for the early stages of the formation of the decomposing communities. Based on the literature cited above, we aimed to gain a mechanistic understanding of how fungal communities formed in senescent leaves are shaped during the early stages of decomposition of *P. australis* litter and how these networks are affected by flooded and dry conditions. Thus, the present study aims to: (i) clarify the effects of the habitat on the diversity and structure of the fungal communities in the decomposing *P. australis* material; (ii) characterise the interactions between the fungal taxa in the communities; and (iii) identify indicator and keystone taxa that can exhibit a substantial impact on fungal diversity during the decomposition process under different ecological conditions.

## 2. Materials and Methods

### 2.1. Site Description

Lake Cerknica is one of the largest intermittent lakes in Europe. It is included as a Natura 2000 site because of its importance for the preservation of endangered birds. The area is flooded for about nine months per year. *Phragmites australis* thrives in areas with different water regimes, at the inner lake area and the edge of the lake along the lake tributaries [3].

### 2.2. Plant Material Collection and Decomposition

For the present study, we collected the lower and the upper leaves of *P. australis* at the end of the vegetation period, when plants were still green, at two different locations at Lake Cerknica, in the riparian zone of the River Stržen (45°43′38.07″ N, 14°24′16.64″ E; 549 a.s.l.) and in the area of the lake at Zadnji Kraj (45°44′27.60″ N, 14°22′12.62″ E; 551 a.s.l.), resulting in four combinations: (1) riparian site, upper leaves; (2) riparian site, lower leaves; (3) lake site, upper leaves; and (4) lake site, lower leaves. The bottom five leaves of the plant were considered as lower leaves, the topmost five fully developed leaves as upper leaves. Plant material was then air-dried at room temperature until constant weight. Litter bags (17 cm × 22 cm, made from 1 mm × 1 mm plastic mesh) were filled with 4 g of material and exposed at a predominantly dry (45°43′38.29″ N, 14°24′16.72″ E) and a predominantly wet (45°43′38.30″ N, 14°24′16.59″ E) site in the vicinity of Gorenje Jezero. The water level measurements were obtained from the nearby hydrological station Gorenje Jezero-Stržen. To limit direct contact with the substrate, the litter bags were fixed to wooden poles. The decomposition phase of the experiment lasted for 45 days, as we wanted to evaluate the importance of fungi inhabiting fresh leaves for the formation of the communities of decomposers. After 45 days, samples were collected and air-dried to constant weight. After drying, plant material was carefully extracted from the litter bags and non-plant material was removed. Subsequently, this air-dried litter was used for the analyses of fungal communities.

### 2.3. Metagenomics

In the current dataset, the whole-community DNA was extracted from com *P. australis* mon reed leaves using the GenElute^®^ Plant Genomic DNA isolation kit (Sigma, St. Louis, MO, USA) following step-by-step procedures from the manufacturer’s manual. Shotgun metagenomic sequencing was done via the Illumina HiSeqX platform (2 × 150 pair-ends) according to the manufacturer’s guidelines using the TruSeq Nano kit (Illumina, San Diego, CA, USA). Analysis and annotation of output data were performed through Metagenomics rapid annotation (MG-RAST) online server [22] with the default parameters. Following quality control (QC), sequences were annotated using BLAT (a BLAST-like alignment tool) algorithm [23] against M5nr and M5rna databases [24], which offers non-redundant integration of numerous databases. MG-RAST pipeline includes two separate pathways—one for rRNA and one for protein coding. In both pathways the sequences are filtered, clustered (90% identity level proteins and 97% identity for rRNA), and identified against databases. In the end, both pathways are combined for annotation step. Fungal profiling yielded 18,325,318 sequences, with 730,000 ± 640,000 sequences per sample for 180 fungal operational taxonomic units (OTUs) at genus level across all samples. The E-value threshold for taxonomic identification was 1 × 10^−5^. The appropriate trophic category (symbiotroph, pathotroph, and saprotroph) for each OTU was inferred by searching against the FUNGuild database [25].

### 2.4. Effects of Habitat on Fungal Communities

Ordination analyses were performed using the phyloseq (v1.34.0) library [26] on a Bray–Curtis distance matrix of rarefied fungal OTU abundances. Only the top 20 most abundant fungal families out of 97, representing 96% of sequences, were included. Diversity indices were calculated using the phyloseq library. Differences between the grouping parameters (decomposition habitat, plant material source location, and leaf age) were tested with permutational analysis of variance (perMANOVA), using *adonis2* function in the vegan (v2.5-7) library [27], followed by the *pairwise.adonis* function (https://github.com/pmartinezarbizu/pairwiseAdonis.git, accessed on 16 December 2021) to assess pairwise differences between the groups. Permutational analysis of multivariate dispersion using the *betadisper* function with 9999 permutations was used to evaluate the perMANOVA results.

### 2.5. Indicator Species

We also performed indicator species analysis to delineate high fidelity differentially abundant patterns. Legendre’s INDVAL procedure was performed in R using the indicspecies (v1.7.9) library [28]. The fungal OTUs with an indicator value > 0.75 and significant *p*-values (*p* < 0.01) were considered as indicator species [29,30]. To avoid random effects caused by rare OTUs, only those with >500 reads were defined as true key players [31] and included in the analysis. Indicator species analysis was run using the multipatt function with 99,999 permutations. *p*-value correction for multiple testing was run using the qvalues function implemented in the qvalue library (v1.38) [32] with a false discovery rate of 5% (q < 0.05). The linear discriminative analysis (LDA) effect size (LEfSe) algorithm using the *ldamarker* function from the microbial (v0.0.17) library [33,34] was used to identify the species that were over-represented in each of the three groups (fresh leaves, dry and wet habitat). The *p*-value cut-off was set to be below 0.001 for the Kruskal–Wallis test and 0.01 for each Wilcoxon test pairwise comparison between the groups. The resulting species with significant differences between the samples were used to build the LDA model and estimate its effect as a discriminant feature. The threshold used to consider discriminative features for the logarithmic LDA score was set to >5.

### 2.6. Co-Occurrence Networks

To remove poorly represented OTUs and reduce network complexity, OTUs with a proportion of the total sequences under 0.02% were removed from the analysis. The co-occurrence network was inferred based on Spearman’s Rho between the pairwise OTUs matrix constructed by the R psych (v2.1.9) library [35]. The *p*-values for multiple testing were calculated using the false discovery rate (FDR) controlling procedure. A valid co-occurrence event was considered robust if correlation coefficient r >|0.6| and if it was statistically significant at *p* < 0.05 [36]. Network images for fresh leaves and each habitat were generated with the igraph (v1.2.7) [34], tidygraph (v1.2) [37], and ggraph (v2.0.5) [38] R libraries. To describe the topological properties of the co-occurrence networks, we calculated the total number of network nodes (representing OTUs), the total number of edges (connections between the nodes representing significant positive correlations between OTUs), average degree, average path length, network diameter, radius, centralisation degree, and density. In addition to the network properties mentioned above, we explored community structure by identifying network modules by utilising the greedy optimisation of the modularity algorithm, as implemented in the R package igraph. For this paper, we interpret co-occurrences or modules as generally representing shared niches rather than necessarily representing direct interactions between organisms [39,40]. We identified keystone OTUs separately for the generated networks and defined them as those nodes within the top 1% of node degree values for each network [41,42].

## 3. Results

### 3.1. Diversity and Community Composition

nMDS ordination showed a good fit (Stress = 0.051) with differences in fungal communities from the fresh leaves and leaves decomposing in the dry or the wet habitat (Figure 1). perMANOVA confirmed that the source location of *P. australis* plants (the geographical location within the lake) and the position of leaves on the plant, and, thus their age, had no effect on the fungal community composition. In contrast, fungal communities from the fresh leaves and leaves decomposing in both habitats (wet and dry) differed one from another (Figure 1). Multivariate dispersion tests were not significant for any of the grouping variables (for habitat = 0.59, for leaf age = 0.65, for *P. australis* source location = 0.42). This confirms that the results are not random and that differences between the fungal communities in individual habitats were driven primarily by the differences in potential colonisers.

Diversity and richness indicators showed lower diversity of fungal communities in leaves decomposing in the wet habitats. In contrast, fresh leaves and leaves decomposing in the dry habitat showed similar diversity/richness of fungal communities (Table 1).

*Phragmites australis* leaves contained at least five fungal phyla, with abundant Ascomycota (95.7%) and Basidiomycota (4.1%). OTUs from other phyla (Chytridiomycota and Glomeromycota) and unassigned phyla represented ~0.1% of all fungal OTUs. Fungi from several orders (Agaricales, Arthoniales, Eurotiales, Helotiales, Hypocreales, Onygenales, Pleosporales, Saccharomycetales, and Sordariales) were observed with high relative abundances in all samples and therefore represent core groups in fungal communities on the *P. australis* leaves (Appendix A).

Most of the fungal genera identified in the *P. australis* leaves were saprotrophs (48.6%), followed by pathotrophs (22.5%), and symbiotrophs (12.6%) (Figure 2). Pure saprotrophs were also the most abundant group of trophic guilds. Pathotroph abundance did not change in the wet habitat compared to the fresh leaves but decreased in the dry habitat. Interestingly, pathotroph-symbiotroph abundance increased in the dry habitat during the decomposition process, whereas in the wet habitat, the same happened with the fungi without a known trophic guild.

### 3.2. Indicator Species

Three indicator taxa (*Exophiala*, *Capnodium*, and *Chrysomyxa*) were found for fresh leaves, with a sensitivity of one, which means a certain probability of finding these species on *P. australis* leaves. In addition, 17 fungal species were indicative for both fresh leaves and dry decomposition habitat. Our analysis yielded no indicator species for the wet habitat. The LEfSe was used to identify the discriminating fungal taxa among the different treatments and identified 16 fungal taxa with significant differences (Figure 3). Five groups were present in a significant share in the fresh leaves: genus *Neosartorya*, family Trichocomaceae, two orders (Eurotiales and Onygenales) and class Eurotiomycetes. Two taxonomic groups were enriched for the dry habitat: genus *Pyrenophora* and family Pleosporaceae. For the wet habitat, we observed nine enriched taxa: two genera (*Sclerotinia* and *Botryotinia*), two families (Sclerotiniaceae and Saccharomycetaceae), two orders (Helotiales and Saccharomycetales), and three classes (Leotiomycetes, Saccharomycetes, and Agaromycetes). These results indicate that many groups at different taxonomic levels can be significantly distinguished between the fresh leaves and the leaves decomposing in the dry or wet habitats. Taken together, LefSe and INDVAL analyses detected a total of four fungal genera (*Capnodium, Chrysomixa*, *Exophiala*, and *Neosartorya*) that were significantly associated with fresh *P. australis* leaves; one indicator genus (*Pyrenophora*) for the leaves decomposing in the wet habitat and two genera for the leaves decomposing in the dry habitat (*Sclerotinia* and *Botryotinia*).

### 3.3. Co-Occurrence Networks

Networks showed a reduced number of edges in the decomposing leaves compared to the fresh leaves (Figure 4, Table 2). Moreover, the network density (density of node connections) was lower for the networks of leaves decomposing in the wet or dry habitats compared to the fresh leaves network. Network diameters, reflecting the maximal distance between two nodes, were lower for the fresh leaves and the wet habitat networks compared to the dry habitat, indicating that these networks are more compact and their nodes are in closer proximity to each other. In contrast, the dry habitat network showed the highest network centralisation measurements, indicating a more decentralised network topology. Habitat-specific networks also varied in the average number of neighbours (average degree). In the fresh leaves and wet habitat network, nodes connected on average with 15.3 and 11.4 nodes, respectively, whereas in the dry habitat, the average was 12.5 nodes. The majority of OTUs in the networks was connected with positive interactions.

Five and ten OTUs were selected as keystone OTUs in the fresh leaves and wet habitat networks, respectively. In contrast, only one keystone OTU was identified in the network of the dry habitat. *Talaromyces* genus was identified as a keystone species in all networks (Appendix A). The other nine keystone OTUs from the decomposing leaves were sensitive to specific habitat conditions (wet). Genera *Talaromyces* and *Uncinocarpus*—the keystone OTU for fresh leaves—were also identified as one of the keystone OTUs in the leaves decomposing in the wet habitat. All the identified keystone taxa showed the ability to degrade cellulose (Cellulolytic Enzyme Database, http://www.microbiome.re.kr/db/celdb/, accessed on 22 November 2021; [43] for *Ajellomyces*, which is not included in the referenced database).

## 4. Discussion

### 4.1. Diversity and Community Composition

In our study the source location of *P. australis* plants (the geographical location within the lake) and the position of leaves on the plant, and, thus their age, had no effect on the fungal community composition. This is contrary to previous research that showed fresh leaves structure, namely its elemental composition, as an important factor affecting the phyllosphere fungal communities [44,45]. Our results show a clear difference between the fungal communities of fresh leaves and the leaves decomposing either in dry or wet habitats. When leaves accumulate on the ground, the rate of microbial colonisation changes significantly [46] due to changes in environmental factors, namely oxygen availability, moisture, temperatures, and biotic interactions that define microbial community development and change significantly [11]. The position of leaves on the plant did not affect the diversity/richness of the fungal community even though these *Phragmites* leaves may have a significantly different elemental composition [47]. The quality of the initial plant material can profoundly shape the developement of the microbial communities of decomposers [44]. Nevertheless, other authors observed that senescence status of the leaves only slightly affected the fungal community [48] and emphasized the dispersal level of the fungal colonisers [49].

*Phragmites australis* leaves in our study contained at least five fungal phyla, with abundant *Ascomycota* and *Basidiomycota*. Somewhat different results were obtained by Van Ryckegem and Verbeken [7], who studied the decomposition process of *P. australis* stems in a tidal marsh. They detected 49 fungal taxa in total; 26 taxa belonged to ascomycetes, 16 taxa to coelomycetes, four taxa to hyphomycetes, and three taxa to basidiomycetes. The phyllosphere fungal communities were primarily composed of Ascomycota and dominated by members of Dothideomycetes, Sordariomycetes, Tremellomycetes, and Eurotiomycetes as previously reported in studies of foliar fungal communities. Other authors observed Dothideomycetes as the dominant group in the phyllosphere fungi [50,51]. However, different dominant taxa were detected in *Fagus sylvatica* [48] and several bryophyte species [52], indicating a different effect of the environment and/or host selection [50,53]. In the present study, additional fungal groups showed similar abundances to Dothiomycetes. This goes especially for Sordariomycetes, which were already observed as a dominant fungal group in the phyllosphere of *P. australis* [54]. Interestingly, basidiomycete *Filobasidiella* (Tremellomycetes) was the most abundant genus on fresh leaves but was replaced with ascomycetes in the decomposing leaves. *Filobasidiella* is often found in the yeast form (*Cryptococcus*) in surface waters as saprobes of insects, plants, and leaf litter [55]. Furthermore, similar to our observations, *Filobasidiella* was among the predominant fungal genera in the non-senescent leaves of the oak *Quercus macrocarpa* [56]. A subsequent decline in *Filobasidiella* and low abundance of basidiomycetes in total is in line with other decomposition studies, which observed ascomycete dominance up to later stages of decomposition [57]. Basidiomycetous species produce enzymes needed to degrade complex polymers [58] and are therefore more important in the late litter decomposition process [59].

Most of the fungal genera identified in the *P. australis* leaves were saprotrophs, although pathotrophs and symbiotrophs were also abundant. These results seem to be in line with the study of Van Ryckegem and Verbeken [6], who observed that primary decomposers possibly develop from several initially endophytic fungi. Many identified fungal species belong to more than one of these ecological categories, which is very common in the fungal kingdom. Notably, several endophytic fungi may be latent pathogens [60] with the selection of their function depending on the environmental parameters. Our results are consistent with a recent study on the fungal endophytes of *Fagus sylvatica* and suggest undeniable advantages of high-throughput sequencing over traditional cultivation in uncovering a good representation of the major functional guilds as well as rare fungal taxa [48].

### 4.2. Indicator Species

We predominantly observed concordance between differential abundance and indicator species analyses in the top-scoring genera. Three indicator species were found for fresh leaves, suggesting that they colonise the still growing or senescent leaves of *P. australis* with a certain probability of finding these species on *P. australis* leaves in the sampled locations. A more significant number of habitats would be needed to confirm these fungi as ubiquitous colonisers of *P. australis* leaves. In addition, a high number of fungal taxa indicative of fresh leaves and the dry habitat suggests that similar communities form on the plants and leaves decomposing at the un-submerged habitats. Among the four indicator genera for the fresh leaves, two genera were identified as saprotrophs *Capnodium* and *Neosartorya,* in addition to pathotroph *Chrysomyxa*, and *Exophiala*. Members of the genus are frequently isolated from natural environments, such as bulk soil, biological crusts, rock surfaces, air, natural water masses, the rhizosphere, and plant tissues [61,62,63]. Such diversity of ecological sources indicates that members of the genus *Exophiala* have versatile lifestyles with adaptations to thrive in multiple habitats, which is especially important in water fluctuating systems such as Lake Cerknica, where habitat conditions frequently change during the vegetation period. *Capnodium*, which was just one out of the several taxa present on the fresh leaves and leaves decomposing in the dry habitat, was also the indicator species for the litter obtained at the Zadnji Kraj (ZK). This suggests minor differences in the initial fungal communities on *P. australis* leaves due to patchy pathogen presence. At this location, *P. australis* is frequently exposed to prolonged dry periods [12] that decrease the vitality of the reed and thus increase its vulnerability to pathogens. The observed community on the fresh leaves suggests that several fungal genera colonise *P. australis* leaves when these are still attached to the plant and endure after the leaves fall off and are subjected to the decomposition process. This is in line with the findings of Koivusaari et al. [64], who observed that foliar microfungi contribute to fungal diversity in the litter.

Interestingly, a shift in indicator species to nectrotrophs was observed in the wet habitat (*Sclerotinia* and *Botryotinia*). Similarly, Wirsel et al. [65] observed *Sclerotinia* on the *Phragmites* material only in flooded sites. Flooding can induce leaf senescence and programmed cell death [66], which is frequently the case in the studied wetland [4]. As preventing early senescence is a resistance strategy of plants against necrotrophs [67], the faster senescence of leaves could speed up nutrient acquisition by necrotrophs and improve their spread in early phases of leaf decomposition.

### 4.3. Co-Occurrence Networks

The co-occurrence network for the fresh leaves showed the lowest fragmentation of the network (the lowest number of modules). Most of the correlations between fungal OTUs in the fresh and decomposing *P. australis* leaves were identified as co-occurrences (positive correlations). The dominance of positive associations suggests that most fungal taxa may act synergistically or share similar ecological niches in the phyllosphere environment [68,69]. A prevalence of positive correlations has also been observed for other microbial networks [70,71], although the proportion was lower for fungal communities in the soil. This suggests that mechanisms of underlying assembly of fungal communities in the soil and phyllosphere are markedly different [53].

Most keystone taxa were exclusive to the individual habitat in which the decomposition process was taking place. We observed nine keystone taxa to be moisture-sensitive, as they were selected for the wet habitat. Three of those had higher abundances and represented common phyllosphere fungi of *P. australis*. The major exception was the genus *Talaromyces*, identified as a keystone taxon in the fresh leaves and leaves decomposing in the wet or dry habitats. *Talaromyces* are known colonisers and decomposers of leaves [72]. These fungi are considered key players in leaf litter decomposition because of their ability to produce a wide range of extracellular enzymes, including those involved in the degradation of cellulose [43].

Furthermore, we observed a positive correspondence between the abundance of the identified keystone taxa and the Chao1 richness indicator. This would suggest that through a high number of positive interactions in the community networks, these keystone taxa have a positive impact on the phyllosphere fungal communities. Keystone taxa are thought to frequently interact with many other taxa, thereby playing an important role in the overall community [36], and our findings certainly support this.

## 5. Conclusions

In conclusion, our results suggest that a complex network of fungi forms is already in the senescent leaves of *P. australis* and persists to the decomposition phase. It seems that habitat has a lower impact on the formation of the community during the early decomposition phase than on the interaction between its different members, as observed from the co-occurrence patterns. Although, we must be careful in interpretations, as co-occurrence networks generally represent shared niches rather than necessarily representing direct interactions between organisms. Nevertheless, the environment can alter the proportions of individual groups in the fungal community due to competition and exploitation of favourable conditions.

## Figures and Tables

**Figure 1 jof-08-00284-f001:**
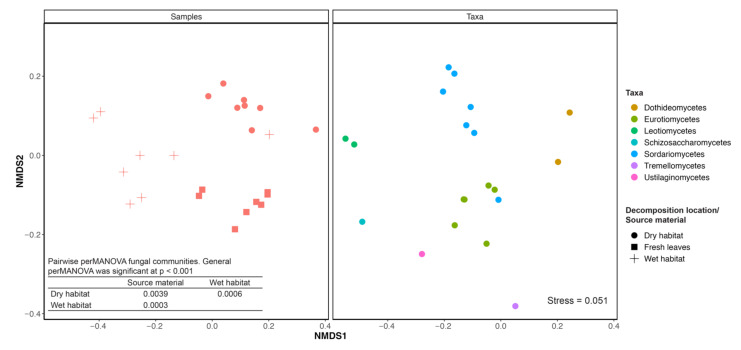
A nonmetric multidimensional scaling (nMDS) ordination for fungal communities from the fresh leaves and *Phragmites australis* leaves decomposing in dry or wet habitat with accompanying stress value. Legend: Different colours represent the taxonomic classes of the top 20 most abundant taxonomic families included in the analysis. Different shapes represent the individual habitats (dry/wet) and fresh leaves. Table represents *p*-values of pairwise perMANOVA for fungal communities in the fresh leaves or leaves decomposing in wet or dry habitat. General perMANOVA was significant at *p* < 0.001.

**Figure 2 jof-08-00284-f002:**
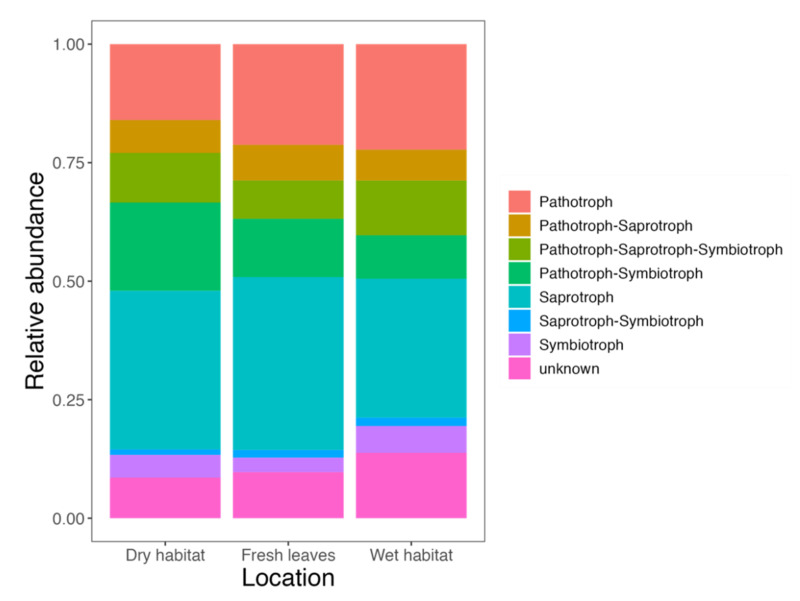
Bar charts showing the trophic guilds of fungal communities in fresh leaves and leaves decomposing in wet or dry habitat. Undefined category combines fungal genera without assigned trophic mode.

**Figure 3 jof-08-00284-f003:**
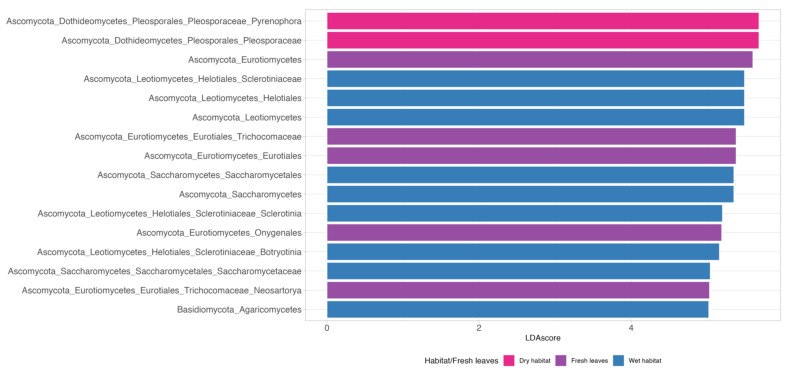
Identification of indicator taxa based on Lefse analysis (LDA scores > 5).

**Figure 4 jof-08-00284-f004:**
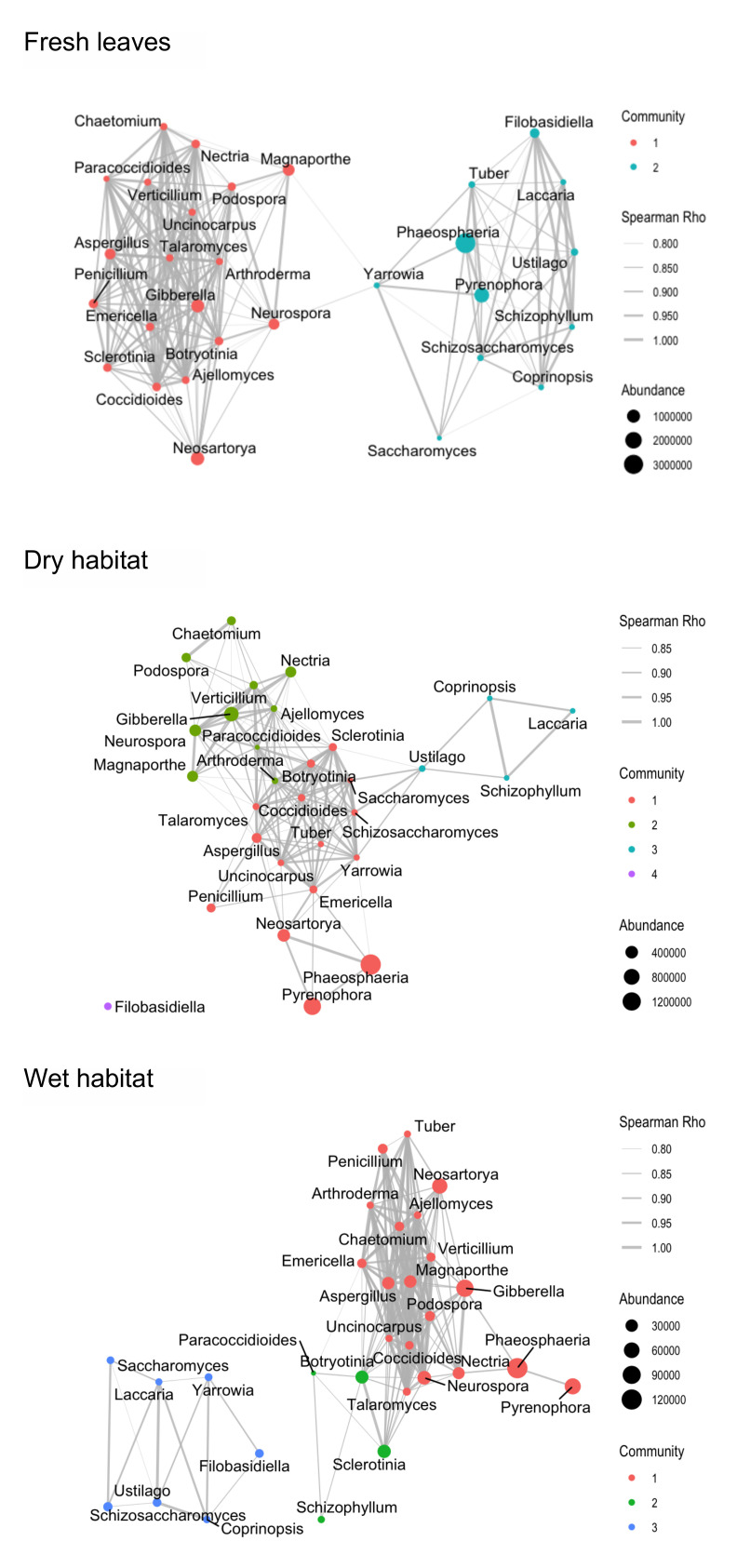
Co-occurrence patterns of fungal communities on fresh leaves and leaves decomposing in the dry and wet habitat visualising significant correlations (ρ > 0.6, *p* < 0.05; indicated with grey lines) between fungal OTUs in leaves of *Phragmites australis*. Different colours of nodes represent individual modules within the community identified using the greedy optimisation of the modularity algorithm. The size of the nodes represents the abundance of sequences for the individual nodes. The weight of the edges represents the Spearman Rho for the individual edge.

**Table 1 jof-08-00284-t001:** Diversity and richness indicators for fungal communities from fresh leaves or leaves decomposing in wet or dry habitat (average ± SD, *n* = 8).

	Observed	Chao1	Shannon	Simpson
Fresh leaves	117.0 ± 3.0	132.4 ± 11.0	2.706 ± 0.08	0.876 ± 0.02
Dry habitat	110.1 ± 6.4	122.9 ± 7.0	2.521 ± 0.14	0.859 ± 0.03
Wet habitat	85.25 ± 9.3	85.25 ± 9.3	2.926 ± 0.26	0.910 ± 0.04

**Table 2 jof-08-00284-t002:** Location-specific network parameters.

Network Parameter	Fresh Leaves	Dry Habitat	Wet Habitat
No. nodes	30	30	30
No. edges	229	187	171
Average degree	15.3	12.5	11.4
Clustering coefficient	0.90	0.72	0.87
Network diameter	4.24	4.43	4.31
Network centralization	0.16	0.29	0.23
Network density	0.53	0.43	0.39
Average path length	2.25	1.96	1.55

## Data Availability

The data that support the findings of this study are available in Likar, Matevz (2022), “Dataset on fungal communities in common reed”, Mendeley Data, V1, doi:10.17632/2mgd5gnjc4.2.

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
