# Peer review of "Original Leaf Colonisers Shape Fungal Decomposer Communities of Phragmites australis in Intermittent Habitats"

_jof, 2022, doi:10.3390/jof8030284_

Round 1

Reviewer 1 Report

I read the manuscript of Likar et al. which describes fungal colonization of leaves of common reed and how it is influenced by 1) origin of the leaves (riparian site vs lake, old vs fresh leaves) and 2) site of colonisation (wet vs dry conditions). I found this study and sampling design interesting and worth of inspecting as common reed is widespread and it produces a lot of biomass.

I think that authors did a good job in presentation of results, the logic behind the analyses is clear and understandable as well as presentation of the results. Graphics is acceptable. I have some minor concerns which are listed below in per-line-comments. However, while most of the methods were well described, the crucial part about sequence processing is lacking the important information about generation of OTUs (clustering algorithm is not mentioned and target gene is not mentioned as well). I wonder what these OTUs actually represent? I understand that shotgun sequencing was used and it produced metagenome which was uploaded to MG-RAST. MG-RAST then predicted genes in metagenome and tried to assign the taxonomy to each gene. Therefore the fungal OTUs here are not based on ITS region but somehow on fungal functional genes? Do I understand correctly? This section in the methods must be clarified. Without the better explanation of the origin of OTUs I can not recommend the article for acceptance. I would like to encourage authors to fill the information and I suggest resubmission after major revision. Please find my per-line comments below.

Lines 46-49 - great reference to previous research

Line 81 - please delete one "intermittent" from the sentence

Line 109 - Please include information about ligation kit which was used before sequencing.

Line 110 - correct 2 x 150 so it includes multiplication symbol

Line 112 - Please specify what was the E-value threshold for taxonomic identification in default parameters as this is important filtering measure.

Line 114 - insert space before ref 24

Line 116 - Instead of range I suggest to report mean +- SD/SE

Line 117 - Please clarify origin of the OTUs, which sequences served for the clustering and what was the clustering radius (97% or similar) as this have impact on further analyses.

Line 126 and also other lines in Methods - Please consider to give full credit to authors of the freeware packages such as vegan which were used. While the tools used are greatly described, their references are missing in the Ref section.

Line 129 - 9,999 permutations

Line 162 - Co-occurrence analysis might be questionable but I like that authors are aware of its pitfalls and included this sentence. Please note that this sentence can also impact conclusion in the final paragraph (lines 372-376). Otherwise graphics of the co-occurrence networks has good quality. 

Line 169 - please correct Stress value so it is in line with the value in the Figure 1.

Line 202-206 - I like this interesting conclusion.

Lines 254-256 - Please consider to specify taxonomy of these OTUs. Maybe paragraph 355-361 might be moved from discussion to results?

Line 262 - To have chemistry data would improve the conclusions. Are these data available. Otherwise explanation of the chemistry effect is sound.

Line 273 - emphasized

Sentence starting at 345 - Could you please explain little bit more why this is the most interesting observation?

Paragraph 355-361 - It contains rather results than discussion.

Line 360 - context of "for Ajellomyces" is unclear, please clarify.

Line 371 - I did not find this result about positive correspondence specifically mentioned in the results. Could you make sure that such result is included in the text?

Line 374 - In possible contrast to 162-164, but we can not say. I prefer to leave the sentence as it is. 

Line 396 - data availability - I could not check data availability because of embargo. Could you please make data public or accessible in reviewer mode?

Figure 1 - Please specify values in the table, are these p-values?

Table 1 - I suggest to report +-SD also in the Observed column. Diversity calculation including Chao1 is not mentioned in the methods, please include it.

References corrections

  • doi missing in references 2, 6, 18, 20, 26, 49
  • please fill paper number for electronic versions of the papers 22, 29, 31, 32, 33, 47, 54, 58

Author Response

Thank you for the opportunity to revise and resubmit this manuscript. I appreciate the time and detail provided by each reviewer and by you. We have incorporated the suggested changes from the copy edited version into the manuscript, reordered the citations and added upper case letters to figures.

Our responses:

I read the manuscript of Likar et al. which describes fungal colonization of leaves of common reed and how it is influenced by 1) origin of the leaves (riparian site vs lake, old vs fresh leaves) and 2) site of colonisation (wet vs dry conditions). I found this study and sampling design interesting and worth of inspecting as common reed is widespread and it produces a lot of biomass.

I think that authors did a good job in presentation of results, the logic behind the analyses is clear and understandable as well as presentation of the results. Graphics is acceptable. I have some minor concerns which are listed below in per-line-comments. However, while most of the methods were well described, the crucial part about sequence processing is lacking the important information about generation of OTUs (clustering algorithm is not mentioned and target gene is not mentioned as well). I wonder what these OTUs actually represent? I understand that shotgun sequencing was used and it produced metagenome which was uploaded to MG-RAST. MG-RAST then predicted genes in metagenome and tried to assign the taxonomy to each gene. Therefore the fungal OTUs here are not based on ITS region but somehow on fungal functional genes? Do I understand correctly? This section in the methods must be clarified. Without the better explanation of the origin of OTUs I can not recommend the article for acceptance. I would like to encourage authors to fill the information and I suggest resubmission after major revision. Please find my per-line comments below.

Our response: MG-RAST pipeline includes two separate pathways – one for rRNA and one for protein-coding. In both pathways, the sequences are filtered, clustered (90% identity level proteins and 97% identity for rRNA), and identified against M5nr and M5rna databases. At the end, both pathways are combined for the annotation step. Additional text was added to the Material and Methods section for further clarification (lines 126-129).

Lines 46-49 - great reference to previous research

Our response: Thank you.

Line 81 - please delete one "intermittent" from the sentence

Our response: We have deleted the unnecessary repetition of the word from the sentence.

Line 109 - Please include information about the ligation kit which was used before sequencing.

Our response: TruSeq Nano kit (Illumina) was used for the preparation of samples for sequencing. We added the appropriate text to the Material and Methods section (line 121).

Line 110 - correct 2 x 150 so it includes multiplication symbol

Our response: Corrected.

Line 112 - Please specify what was the E-value threshold for taxonomic identification in default parameters as this is important filtering measure.

Our response: The R threshold was 1e-5. We added the appropriate text to the Material and Methods section (lines 131-132).

Line 114 - insert space before ref 24

Our response: Corrected.

Line 116 - Instead of range I suggest to report mean +- SD/SE

Our response: Corrected.

Line 117 - Please clarify origin of the OTUs, which sequences served for the clustering and what was the clustering radius (97% or similar) as this have impact on further analyses.

Our response: Additional text was added to the Material and Methods section (lines 126-129) for clarification.

Line 126 and also other lines in Methods - Please consider to give full credit to authors of the freeware packages such as vegan which were used. While the tools used are greatly described, their references are missing in the Ref section.

Our response: As suggested, we have added references for all used R libraries to the References section.

Line 129 - 9,999 permutations

Our response: Corrected.

Line 162 - Co-occurrence analysis might be questionable but I like that authors are aware of its pitfalls and included this sentence. Please note that this sentence can also impact conclusion in the final paragraph (lines 372-376). Otherwise graphics of the co-occurrence networks has good quality.

Our response: An additional sentence was added to the Conclusions section to alert the readers to the pitfall already mentioned in the Materials and Methods section (lines 442-444).

Line 169 - please correct Stress value so it is in line with the value in the Figure 1.

Our response: Corrected.

Line 202-206 - I like this interesting conclusion.

Our response: Thank you.

Lines 254-256 - Please consider to specify taxonomy of these OTUs. Maybe paragraph 355-361 might be moved from discussion to results?

Our response: Paragraph 355-361 was moved to the Results section, as suggested (lines 298-305). The OTUs’ information is now referenced through Suppl. Table S1.

Line 262 - To have chemistry data would improve the conclusions. Are these data available. Otherwise explanation of the chemistry effect is sound.

Our response: The chemistry data was not measured.

Line 273 - emphasized

Our response: Corrected.

Sentence starting at 345 - Could you please explain little bit more why this is the most interesting observation?

Our response: We removed the word and rewrote the sentence, as we did not want to emphasize this part of the discussion in comparison with the others.

Paragraph 355-361 - It contains rather results than discussion.

Our response: The paragraph was moved to the Results section, as suggested (lines 298-305).

Line 360 - context of "for Ajellomyces" is unclear, please clarify.

Our response: Ajellomyces was not included in the referenced database, therefore we included an additional reference for this genus. We added additional text for explanation (lines 285-286).

Line 371 - I did not find this result about positive correspondence specifically mentioned in the results. Could you make sure that such result is included in the text?

Our response: Additional text was added to the Results section (lines 263-264).

Line 374 - In possible contrast to 162-164, but we can not say. I prefer to leave the sentence as it is.

Our response: As suggested, we have left the sentence as it is.

Line 396 - data availability - I could not check data availability because of embargo. Could you please make data public or accessible in reviewer mode?

Our response: The data for a review is available through a direct link: https://data.mendeley.com/datasets/2mgd5gnjc4/draft?a=96b444be-aa4a-4f69

Figure 1 - Please specify values in the table, are these p-values?

Our response: The caption for Figure 1 was corrected and explanation of the values (p-values) in the table was added.

Table 1 - I suggest to report +-SD also in the Observed column. Diversity calculation including Chao1 is not mentioned in the methods, please include it.

Our response: We have added the +/-SD values for the observed column as suggested. Diversity indices were calculated using the phyloseq library. We added this information to the Material and Methods section (lines 138-139).

References corrections

  • doi missing in references 2, 6, 18, 20, 26, 49

Our response: We have added the missing DOI numbers to the references. References 6, 18 and 49 (not 58) don’t have a DOI number.

  • please fill paper number for electronic versions of the papers 22, 29, 31, 32, 33, 47, 54, 58

Our response: We have added the paper numbers to the references.

Reviewer 2 Report

The article "Original leaf colonisers shape fungal decomposer communities of common reed in intermittent habitats" contains interesting data about mycobiota on Phragmites australis  at various stages of decomposition. In general, the study was carried out at a good methodological level, the results are well described and illustrated. In the presented form, there are no fundamental remarks to the article. However, authors should check the italicization of all fungal taxa, and also, it seems to me, should replace Common reed in most cases with the Latin name of the species Phragmites australis.
As a suggestion for future work, the study would be much more informative if the authors also used cultural isolation techniques to identify the most characteristic, dominant, and interesting species of fungi.

Author Response

Thank you for the opportunity to revise and resubmit this manuscript. I appreciate the time and detail provided by each reviewer and by you. We have incorporated the suggested changes into the manuscript.

Our responses:

The article "Original leaf colonisers shape fungal decomposer communities of common reed in intermittent habitats" contains interesting data about mycobiota on Phragmites australis  at various stages of decomposition. In general, the study was carried out at a good methodological level, the results are well described and illustrated. In the presented form, there are no fundamental remarks to the article.

However, authors should check the italicization of all fungal taxa, and also, it seems to me, should replace Common reed in most cases with the Latin name of the species Phragmites australis.

Our response: We have rechecked the manuscript and italicized any unitalicized fungal genera. We also replaced common reed with Phragmites australis throughout the document.

As a suggestion for future work, the study would be much more informative if the authors also used cultural isolation techniques to identify the most characteristic, dominant, and interesting species of fungi.

Our response: Thank you for this suggestion. We will include it in the plans for future analyses.

Round 2

Reviewer 1 Report

Dear Authors,

thank you for considering my suggestions, I am satisfied with the details added to the manuscript.

Please could you share again the link to deposited data so I am able to verify them? The current link provided during review does not work and data are under embargo.

Author Response

Thank you for the opportunity to revise and resubmit this manuscript.

We have generated a new shared link, just in case the old one was corrupted: https://data.mendeley.com/datasets/2mgd5gnjc4/draft?a=96b444be-aa4a-4f69-8623-12e98fea3352

Additionally, for this review, we uploaded the dataset shared on Mendeley Data also as an Excel file to Dropbox. Link to the dataset: https://www.dropbox.com/s/mt9m6j5qqvkk23z/shared_data.xlsx?dl=0

Kind regards,

Matevz Likar